# Safety of Non-Vitamin K Antagonist Oral Anticoagulant Treatment in Patients with Chronic Kidney Disease and Kidney Transplant Recipients

Mikołaj Młyński [1] , Mikołaj Sajek [1], Zbigniew Heleniak [2],* and Alicja Dębska-Ślizień [2]

1   Medical Faculty, Medical University of Gdańsk, 80-210 Gdańsk, Poland;
    mikolaj.mlynski@gumed.edu.pl (M.M.); sajekmikolaj@gumed.edu.pl (M.S.)
2   Department of Nephrology, Transplantology and Internal Medicine, Medical University of Gdańsk,
    80-210 Gdańsk, Poland; adeb@gumed.edu.pl
*   Correspondence: zbigniew.heleniak@gumed.edu.pl

**Abstract:** The use of novel oral anticoagulants in patients with impaired renal function or undergoing immunosuppressive therapy is limited due to the risk of drug-to-drug interactions and anticoagulation-related adverse events. This article aims to assess the current data on the safety of direct-acting oral anticoagulant-based therapy in the population of kidney transplant recipients and patients with impaired renal function. The most important factors affecting the safety of treatment are the incidence of bleeding events, thromboembolic events, deaths and drug-to-drug interactions. The available data were compared to the findings on warfarin-based anticoagulation. Findings on the use of novel oral anticoagulants in kidney transplant recipients are limited yet promising in terms of safety and efficacy of use. However, current recommendations state that the co-administration of non-vitamin K antagonist oral anticoagulants with several immunosuppressive agents is contraindicated.

**Keywords:** chronic kidney disease; kidney transplant recipient; anticoagulation; novel oral anticoagulants

## 1. Introduction

Novel oral anticoagulants (NOACs) have been commonly used in patients facing a higher risk of cardiovascular (CV) incidents such as atrial fibrillation (AF) and venous thrombo-embolic events (VTE), successfully decreasing numbers of CV incidents in the general population. The probability of those incidents increases in specific groups of patients, including those with chronic kidney disease (CKD) where the risk of developing AF averages between 19–24%, reaching up to 27% in patients with end-stage kidney disease (ESKD). In kidney transplant recipients (KTRs) it reaches 7.3% within 36 months from the procedure in comparison to 2% in the general population [1–3]. In this article, we are going to focus on the use of NOACs in post-transplant treatment and compare them with the commonly used vitamin K antagonists (VKA), primarily warfarin, in terms of effective anticoagulation and safety of use, including the risk of bleeding, incidence of thromboembolic events and drug-to-drug interactions (DDI). The use of NOACs in KTRs is not strictly contraindicated but requires renal function assessment and therapeutic drug monitoring of immunosuppressive drugs, considering their predominant renal excretion and possible pharmacokinetic interactions between the two groups [4]. NOACs are successfully used as a prevention of thromboembolic events in patients with AF and treatment of such as deep vein thrombosis (DVT) and pulmonary embolism (PE) in the general population as they have a wider therapeutic window, lower frequency of intracranial bleeding and do not require routine monitoring. Atrial fibrillation increases the risk of ischemic stroke 5-fold [3] and is found in 24% of patients during the acute phase of ischemic stroke. Stroke risk is 43% greater in patients suffering from CKD without atrial fibrillation in comparison to the general population [5]. Among patients with CKD, AF increases the risk of stroke between

1.5-fold and 2.5-fold depending on the stage of kidney dysfunction and albuminuria [6]. History of atrial fibrillation among KTRs is associated with an increased risk of graft failure and post-transplant mortality, as well as a 37% higher risk of ischemic stroke [7]. Increased risk of cerebrovascular events among patients with CKD and KTRs with or without atrial fibrillation indicates the necessity of proper and safe anticoagulation therapy in the above groups of patients.

## 2. Oral Anticoagulants

### 2.1. Warfarin

Warfarin, a commonly used vitamin K antagonist (VKA), has been proven to decrease the risk of ischemic stroke for both patients with CKD and KTRs. Warfarin-based anticoagulation among patients with CKD provided a decrease in stroke risk from 26% to 9% [8], which is comparable with the relative reduction in stroke risk in the general population [9], while KTR studies have shown a trend towards a decrease in composite endpoints of death, stroke and gastrointestinal bleeding [10]. On the other hand, warfarin-based anticoagulant therapy significantly enhances the risk of major bleeding events. Studies show that the safety of warfarin use strongly depends on maintaining the INR level within the therapeutic norms (2–3) without presenting strong dependence on the GFR level. In a prospective cohort study of 1273 long-time warfarin users it was demonstrated that compared with patients with a GFR of >60 mL/min per 1.73 $m^2$, those with a GFR of 30–44 mL/min per 1.73 $m^2$ and those with a GFR < 30 mL/min per 1.73 $m^2$ had 2.2-fold and 5.8-fold higher risks, respectively, of major bleeding events at an INR value > 4, but the same study showed that GFR did not modify the risk of hemorrhage for INR values < 4 [11]. Importantly, KTRs receiving warfarin-based OAC require stricter monitoring and lower doses of anticoagulants in order to maintain the safety of treatment [12].

For many years the world has not been presented with an alternative oral anticoagulant, until 2010, when the Food and Drugs Administration approved dabigatran as a new option for the therapy. Since then, three other oral agents: rivaroxaban, apixaban and edoxaban have been approved. Even though the pharmacokinetic features of NOACs are dependent on renal clearance to some degree, they have been commonly prescribed for patients with kidney diseases suffering from atrial fibrillation and increased thromboembolic event risks with good effects and a low risk of bleeding (Table 1).

**Table 1.** Essential clinical information on the use of NOACs.

|  | Apixaban | Rivaroxaban | Dabigatran |
|---|---|---|---|
| Mechanism of action | Direct factor Xa inhibition | Direct factor Xa inhibition | Direct factor IIa inhibition |
| Indications | Prevention and treatment of VTE Prevention of stroke in individuals suffering from NVAF | Prevention and treatment of VTE Prevention of CV incidents in ASCVDPrevention of stroke and systemic embolism in individuals suffering from NVAF | Prevention and treatment of VTE Prevention of stroke and systemic embolism in individuals suffering from NVAF |
| Standard dosage | 5 mg orally twice a day * | 20 mg orally once a day | 150 mg orally twice a day |
| Dosage in CKD | 2.5 mg orally twice a day (in CrCl 15–30 mL/min) | 15 mg orally once a day (in CrCl 15–50 mL/min) | 110 mg orally twice a day ** |
| Minimal CrCl at which the drug is administered | Not recommended for patients with CrCl < 15 mL/min | Not recommended for patients with CrCl < 15 mL/min | Not recommended for patients with CrCl < 30 mL/min |
| Therapeutic effect monitoring | Anti-Xa activity | Anti-Xa activity | TT/dTT (and aPTT) |
| Antidote *** | Recombinant modified human factor Xa—andexanet alfa | Recombinant modified human factor Xa—andexanet alfa | Monoclonal antibody against dabigatran—idarucizumab **** |

* Decreased dose of 2.5 mg orally twice a day is administered in patients with two out of three of the following criteria: age ≥ 80 years, body weight ≤ 60 kg, creatinine concentration ≥ 1.5 mg/dL. ** As well as in patients with a high risk of bleeding. *** Intended uses of NOAC activity reversing drugs include: unplanned emergency surgeries or procedures and uncontrolled life-threatening bleeding episodes. **** Recommended dose of idarucizumab is 5 g—the dose may be repeated in case of recurrence of bleeding or indications for a second emergency procedure. No dose adjustments are required for patients with renal impairment.

## 2.2. Apixaban

Apixaban is a direct Xa factor inhibitor indicated to prevent and treat VTE and decrease the risk of a stroke in individuals suffering from non-valvular atrial fibrillation (NVAF). In the Apixaban Versus Acetylsalicylic Acid to Prevent Stroke in Atrial Fibrillation Patients Who Have Failed or Are Unsuitable for Vitamin K Antagonist Treatment (AVERROES) trial ($n$ = 5.599 subjects), patients with AF were randomized to apixaban 5 mg twice a day and aspirin. A reduced dose of apixaban 2.5 mg twice per day was given to patients who met at least two of the following criteria: serum creatinine between 1.5 and 2.5 mg/dl, age $\geq$ 80 years, and body weight $\leq$ 60 kg. The study indicated apixaban's superiority over aspirin in preventing stroke and systemic embolization [13]. Open-label extension of the study presented data supporting the safety and efficacy in the long-term use of apixaban in patients suffering from AF [14]. In the Apixaban for Reduction in Stroke and Other Thromboembolic Events in Atrial Fibrillation study (ARISTOTLE), apixaban has been shown to be superior to warfarin in preventing stroke and systemic embolism, while also reducing the risk of bleeding and mortality [15]. The ARISTOTLE study showed that in CKD stage G4, apixaban might reduce the risk of stroke (1.27 vs. 1.6% per year) and major bleeding (2.13 vs. 3.09% per year) in comparison with warfarin during a median follow-up of 1.8 years; however, the study only included a small number of participants in this CKD stage (3%, 270 individuals) [15,16].

Pharmacodynamic features of apixaban result from factor Xa inhibition and even though the regular control of an anticoagulant effect is most often unnecessary during therapy, major changes in screening plasma coagulation tests (mostly APTT and PT) have been observed. However, neither APTT nor PT can be used for the laboratory control of NOAC activity in order to adjust the dose and ensure that a therapeutic effect is obtained. Studies showed different results of screening plasma coagulation tests in various individuals receiving the same dose of NOAC therapy [17,18]. Furthermore, Apixaban, along with other NOACs, exerts anti-Xa activity, which can be used to assess the concentration of the drug (Rotachrom Heparin anti-Xa assay). Such dose adjustments are however inadvisable since the therapeutic range of NOACs remains unknown [19]. Nevertheless, as stated by Baglin et al., in the recommendation on measuring oral direct inhibitors of thrombin and factor Xa, it is advised to monitor the anticoagulant effect in specific clinical scenarios, such as bleeding incidents, prior to surgery and perioperatively, in patients who take other drugs which may cause pharmacokinetic interactions or who may benefit from dose adjustments due to extreme body weight. Other indications include patients with decreased renal function, suspicion of overdose, necessity for reversal of anticoagulation and compliance monitoring [20].

Apixaban's absorption occurs primarily in the upper gastrointestinal tract reaching maximal plasma concentration (Cmax) usually after 3 to 4 h. It is bound by plasma proteins in ~93% of healthy subjects, which is comparable to patients with ESRD. Absolute bioactivity of the drug is approximately 50% and it indicates dose-proportional increases in AUC for oral doses up to 10mg. In humans, apixaban is approximately 87% bound to plasma proteins and the volume of distribution (Vss) is approximately 21L. Apixaban is mainly metabolized by CYP3A4/5; however, in human plasma, it occurs mainly in the unchanged form and studies have not detected any active metabolites in the bloodstream. The renal excretion of apixaban accounts for approximately 27% of the total body clearance, followed by 50% eliminated through biliary and intestinal secretion into the feces [17]. Based on low-quality evidence, apixaban appears to be the preferred agent in patients with renal insufficiency, but further studies are warranted [21].

## 2.3. Rivaroxaban

Rivaroxaban is also a direct Xa inhibitor indicated in the prevention of VTE and CV incidents in atherosclerotic cardiovascular disease (ASCVD), prevention of stroke and systemic embolism in NVAF and treatment of VTE. In ROCKET-AF (Rivaroxaban Once Daily Oral Direct Factor Xa Inhibition Compared with Vitamin K Antagonism for

Prevention of Stroke and Embolism Trial in Atrial Fibrillation), a randomized controlled trial of 14,264 patients with AF at a moderate to high risk for stroke (mean CHADS2 score $3.5 \pm 0.9$), all of whom had CrCl > 30 mL/min, rivaroxaban had similar effects to warfarin in preventing stroke and systemic embolism [22]. Moreover, according to the study, major bleeding occurrence was not significantly different between the two groups. The FDA approved a 20 mg once-daily rivaroxaban dose in November 2011. The label consisted of recommendations of 15 mg once-daily dose for patients with CrCl between 15 and 50 mL/min. Rivaroxaban is not recommended for use in patients with CrCl < 15 mL/min or those undergoing dialysis.

Absolute bioavailability is dose dependent where almost complete absorption (80 to 100%) is achieved at the 10 mg dose but reduced to 66% for the 20 mg dose. Absorption occurs primarily in the proximal small intestine with peak concentrations observed 2 to 4 h following oral intake. In humans, rivaroxaban is extensively bound to plasma proteins, approximately 92–95%, mainly albumin. The volume of distribution is moderate with a steady-state volume of distribution (Vss) of approximately 50 liters. Approximately 2/3 of the rivaroxaban dose is metabolized, half of which is eliminated via the kidneys and the other half via the feces. The remaining 1/3 of the administered rivaroxaban dose is excreted by the kidneys in the urine in an unchanged form mainly through active renal secretion. Unchanged rivaroxaban is the most important compound in human plasma; no active circulating metabolite is present. The elimination of rivaroxaban from the plasma occurs with a terminal half-life of 5 to 9 h in young individuals and a terminal half-life of 11 to 13 h in the elderly. Rivaroxaban affects the clotting times by, most importantly, prolonging the PT and aPTT, the latter with a curvilinear concentration–response relationship. aPTT is prolonged 1.5- to 2-fold at peak plasma concentration with normalization all the way through. However, in patients with CrCl < 50 mL/min, the measurements appear to be less sensitive. PT measurement can be used to determine the approximate degree of anticoagulation, with normal PT corresponding with its unsatisfying level. Measurement of plasma rivaroxaban concentration may be useful in situations where DOAC-related bleeding must be excluded and the contribution of oral anticoagulants to the bleeding event must be assessed. In patients requiring an emergency surgical intervention the plasma drug level measurement may also be useful [23].

### 2.4. Dabigatran

Dabigatran is a direct thrombin (factor IIa) inhibitor also recommended in the primary prevention and treatment of VTE and prevention of stroke and systemic embolism in NVAF. According to the RE-LY study, dabigatran-based treatment in patients with NVAF, in comparison to warfarin, has been associated with a noninferior reduction in stroke risk with a lower risk of bleeding, while administered in doses of 110 mg twice a day, and a superior reduction in stroke risk, but a similar risk of bleeding while being administered in doses of 150 mg twice a day [24]. Moreover, studies on the pharmacokinetic profile of dabigatran have shown that, in patients with CrCl of 15–30 mL/min, a twice daily regimen of 75 mg dose could be applied, achieving a similar exposure to the drug. However, patients with even moderate renal dysfunction (CrCl 50–80 mL/min) who converted from warfarin to dabigatran have a three times higher risk of bleeding in comparison to patients who did not undergo treatment conversion [25]. Dabigatran is administered orally in the form of a prodrug, dabigatran etexilate, with a mean bioavailability of 6.5%. It is completely converted by nonspecific hydrolases to the active product, reaching peak concentration at around 1.5–3 h after administration—the distribution volume equals 50–70 L. Dabigatran undergoes rapid distribution within the body tissues, resulting in a rapid decrease in plasma concentration to <30% Cmax within 4–6 h from administration, followed by an elimination phase. The plasma half-life of dabigatran averages 12–14 h and is dose independent. Studies with radiolabeled dabigatran show that ~35% of the drug is bound to plasma proteins. Importantly, dabigatran etexilate is not metabolized by CYP enzymes and does not induce or inhibit their activity [26,27]. Elimination of

dabigatran occurs mostly through the kidneys (80%) in an unchanged form, while 20% of the drug is conjugated by glucuronosyltransferases to pharmacologically active glucuronide compounds, which remain pharmacologically active and can be detected in urine [26,27]. There is a dose-dependent effect of dabigatran on laboratory clotting tests. The aPTT is sensitive to dabigatran, showing a curvilinear concentration–response relationship, with a steep increase at low concentrations and linearity above a dabigatran concentration of 400 ng/mL. In patients taking 150 mg twice daily, the median peak aPTT ratio is approximately two times higher than the peak observed in controls, and the median trough aPTT is 1.5 times higher than median observed in the controls [28]. Above 100 ng/mL, the aPTT is substantially prolonged [29]. As said before, the aPTT is useful as an easily available method for determining the relative degree of anticoagulation but should not be used to determine the plasma drug level. On the other hand, in comparison to aPTT, PT is insensitive to dabigatran. At 100 ng/mL, the PT is usually within the normal range. [29] Similarly to aPTT, TT is sensitive to the antithrombin effect of dabigatran. The sensitivity of this parameter is presenting itself with a linear concentration–response relationship over the therapeutic range of the drug. A normal TT correlates with subtherapeutic (low or even undetectable plasma concentration of dabigatran [20,28,29]. Moreover, in comparison to aPTT, the measurement of TT and diluted thrombin time (dTT) is more sensitive at lower plasma dabigatran concentrations—normal TT/dTT is not observed at therapeutic plasma drug concentrations in contradt to aPTT [30,31].

Although the monitoring of NOAC-based treatment could be based on plasma drug concentration, therapeutic ranges have not yet been set. However, in the studies focused on pharmacokinetics and pharmacodynamics of NOACs, it was able to determine that administering fixed doses of the drugs corresponds to stable and predictable clinical effect, even though the plasma concentration ranges were wide [32,33]. As the novel oral anticoagulants show a stable profile of therapeutic outcomes, it is necessary to determine how the profile changes in specific populations, such as KTRs.

## 3. Immunosuppressive Agents

The most commonly used immunosuppressive agents in the maintenance of graft function in KTRs are calcineurin inhibitors (cyclosporine and tacrolimus), sirolimus, mofetil mycophenolate (MMF) and CTLA-4 fusion proteins, such as belatacept.

In these groups of immunosuppressants, the most significant DDIs have been reported during the administration of calcineurin inhibitors (Table 2). Tacrolimus and cyclosporine have been shown to have the potential to inhibit enzymes responsible for drug metabolism and efflux transporters, e.g., CYP450 3A4 isoenzyme or efflux transporters P-glycoprotein (P-gp) and breast cancer resistance protein (BCRP).

**Table 2.** Mechanism of action and possible mechanism for DDIs in immunosuppressive agents.

|  | Cyclosporine [34] | Tacrolimus [35] | Sirolimus [36] | Mofetil Mycophenolate [37] | Belatacept [38] |
|---|---|---|---|---|---|
| Mechanism of action | Inhibition of calcineurin and NF-kB pathway, leading to decreased IL-2 production; Inhibition of T-cell activation | Inhibition of calcineurin and NF-kB pathway, leading to decreased IL-2 production; Inhibition of T-cell activation | Inhibition of mTOR protein kinase pathway; Inhibition of T-cell and B-cell activation | Inhibition of inosine monophosphate dehydrogenase; Inhibition of DNA synthesis in lymphocytes | Selective inhibition of T-cell co-stimulation by antigen presenting cells |
| Possible mechanism of DDI | Inhibition of CYP3A4, P-gp and BCRP | Inhibition of CYP3A4 and P-gp | No interactions with NOACs reported | DDIs mainly related to absorption of the drug, no interactions with NOACs reported | No formal DDIs reported |

## 4. Drug-to-Drug Interactions

In this article, our aim is to mostly focus on pharmacokinetic interactions with immunosuppressive drugs used in KTRs, as the co-administration of NOACs and immunosuppressive drugs is continuously a subject of discussion among clinicians.

### 4.1. Apixaban

Apixaban is metabolized by cytochrome isoenzymes, mainly CYP3A4/5, but also CYP1A2, CYP2C8, CYP2C9, CYP2C19 and CYP2J2. With only 27% of total plasma clearance of apixaban taking place in the kidneys, renal dysfunction is a factor significantly contributing to the potential increase in drug exposure. In in vitro models, apixaban did not affect the activity of cytochrome isoenzymes or P-gp transport of other drugs. However, being the substrate for both CYP3A4/5, P-gp and also BCRP (with predominant BCRP-mediated efflux) it is justified to consider the influence of co-administration of potential inhibitors of said proteins on apixaban exposure [39,40].

Apixaban exposure is affected mostly by calcineurin inhibitors (CNI), with surprisingly opposite effects on AUC and Cmax. Co-administration of cyclosporine increased AUC and Cmax by 20 and 43%, respectively, while tacrolimus decreased them by 22% and 13%, respectively. Interactions requiring dose adjustment are described mostly regarding strong CYP3A4/5 and P-gp inhibitors and inducers. Strong dual inhibition of cytochrome isoenzymes metabolism and P-gp efflux may result in an up to 2-fold increase in apixaban exposure in patients with decreased renal function. Hence, a 50% dose reduction is recommended when apixaban is co-administered with, e.g., ketoconazole and other azole antimycotics. However, dose adjustments are not recommended in patients taking moderate and weak CYP3A4/5 and P-gp inhibitors, as the AUC and Cmax increases were not clinically significant [15]. Yet, according to the 2021 EHRA Guidelines on the use of NOACs, apixaban-based treatment should be conducted with caution then co-administered with cyclosporine and/or sirolimus, while co-administration with tacrolimus should be avoided [31].

### 4.2. Rivaroxaban

The major metabolic pathways affecting rivaroxaban clearance are the hepatic metabolism (predominantly CYP3A4/5 and CYP2J2 isoenzymes) responsible for over 50% of drug elimination and renal clearance (including active efflux by P-gp and BCRP) responsible for ~36% of drug elimination. Rivaroxaban exposure has proved to be affected by both cytochrome isoenzyme and efflux protein inhibitor. Yet, rivaroxaban is not an inhibitor or inductor of said metabolic pathways itself. Thus, DDI regarding this factor Xa inhibitor should be considered mostly in terms of rivaroxaban exposure [39].

Recommendations regarding rivaroxaban modifications have been created mainly for the co-administration of drugs being strong inhibitors of CYP3A4/5 and P-gp, in which case the use of rivaroxaban should be avoided. Considering the DDIs with substances being moderate and weak inhibitors, clinically significant rivaroxaban exposure increase has been reported mostly when more than one DDI occurred. Although pharmacokinetic studies focusing on interactions with CNI are limited, one research study showed a strong increase in rivaroxaban concentration when co-administered with cyclosporine but not tacrolimus [41,42]. However, rivaroxaban-based treatment was not recommended by EHRA for patients taking tacrolimus and should be carefully controlled in patients taking cyclosporine and sirolimus—similarly to apixaban [31].

### 4.3. Dabigatran

According to studies conducted in vitro, dabigatran etexilate and dabigatran are not interacting with CYP isoenzymes, as co-administration with CYP3A4, CYP2C9 and CYP2C19 inhibitors has not shown any significant change in the distribution of dabigatran [27]. However, dabigatran etexilate is a substrate for P-gp, so any changes in drug absorption should be monitored. Studies with the co-administration of dabigatran etexilate and P-gp

inhibitors show varying alteration in Cmax, up to a 50–60% increase while co-administered with amiodarone [27,39].

Considering the possible DDI of dabigatran and dabigatran etexilate with immunosuppressive agents, the inhibition of P-gp is the most significant factor affecting the safety of dabigatran-based anticoagulation. In in vitro studies, cyclosporine strongly inhibits the efflux of dabigatran, up to 83%, which is mainly attributed to the inhibition of intestinal efflux. According to the U.S. Food and Drug Agency, co-administration of dabigatran etexilate and CNI is safe in patients with CrCl > 50 mL/min, as the risk of high dabigatran exposures is low. In patients with decreased renal function, dose reduction is recommended in prophylaxis of stroke and systemic embolism in patients with NVAF, while it is contraindicated in the treatment of VTE. As reported by the European Medicines Agency, concomitant treatment with dabigatran etexilate and P-gp such as CNI is generally contraindicated [43,44]. The aforementioned reports are supported by the guidelines published by EHRA, which state that dabigatran etexilate should not be co-administered with cyclosporine, but also tacrolimus [31].

### 5. Drug-to-Food Interactions and Interactions with Dietary Supplements

In a patient population such as KTRs, it is crucial to take into consideration other sources of pharmacokinetic and pharmacodynamic interactions with NOACs, as the co-administration of more agents affecting the same routes of drug absorption and elimination may affect the plasma drug concentration to a greater extent than in closely monitored trials.

Inspecting the relationship between food intake and NOAC administration, we can observe that dabigatran etexilate and apixaban can be safely taken with or without food, with no impact on the drug absorption process. However, rivaroxaban absorption, most importantly when taken in doses of 15 mg and more, is affected by food intake. Administration of the drug without food decreases its absorption by up to 34%, which may lead to subtherapeutic plasma drug concentration levels [45].

However, it has been reported that certain dietary supplements and herbal products can interact with NOAC absorption, metabolism and elimination. In a review written by Grześk and colleagues [43], the authors present a list of potential chemical compounds, which may interact with novel oral anticoagulants, together their natural food and herbal sources. These substances act mostly by the induction or inhibition of cytochrome P450 isoenzymes, P-gp or BCRP, although few exhibit antiplatelet activity. The best examined herb is St. John's Wort, whose co-administration with NOACs significantly decreases the plasma drug concentration, what is also mentioned in the EHRA Practical Guide on the Use of Non-Vitamin K Antagonist Oral Anticoagulants in Patients with Atrial Fibrillation [31]. More sources of substances of interaction-causing potential have not yet been examined in studies on humans.

### 6. Safety

As with the use of other anticoagulants, patients taking NOACs should be carefully monitored for signs of bleeding though they do not appear to significantly increase the risk of bleeding events. Safety of use of novel oral anticoagulant factors in patients with CKD or those who underwent kidney transplantation is not well studied. In a retrospective study by scientists from Tafi University, among 42 patients after KT (duration since surgery: 8.8 months ± 7.4) they observed three bleeding events (7.1%) consisting of one patient treated with rivaroxaban 15 mg daily and two patients who received 2.5 mg apixaban twice daily. The patient who took rivaroxaban developed postoperative intraocular bleeding one day after an ophthalmologic procedure, but it was assumed to be unrelated to the medication. The other two bleeding events were minor per-rectum bleeding that did not require surgical intervention or cessation of NOACs. Apart from that, no thromboembolic events were observed in the study [4]. In a retrospective study conducted by Wallvik, Renlund et al., a total of 18,219 patients with NOAC-treated deep vein thrombosis (DVT) or pulmonary embolism (PE) were included. The majority had a venous thromboembolic

event for the first time (85.6%). The distribution of NOAC substances were rivaroxaban 54.8%, apixaban 42.0%, dabigatran 3.2% and edoxaban 0.1%. In total, 13.2% of the patients used low dose NOAC (i.e., rivaroxaban 10 mg once a day, apixaban 2.5 mg twice a day, dabigatran 110 mg twice a day or edoxaban 30 mg once a day). The median follow-up time was 183 days. Among the 18,219 patients, 938 had a major bleeding and the rate of major bleeding was 6.62 (95% CI 6.19–7.06) per 100 treatment years. Renal failure was associated with a higher risk of major bleeding in this study, but the finding was not confirmed in multivariate analysis [46]. Another study, conducted by Bixby, Shaikh et al., compared 197 adult KTRs who received either warfarin or NOACs due to VTE or AF. No statistically significant difference in major bleeding was shown between the warfarin-treated and NOAC-treated individuals. Importantly, a comparison of patients receiving warfarin and apixaban showed a lower incidence of major bleeding events in the apixaban-treated patients (6.7 vs. 21.4%, $p = 0.014$) and a trend towards lower composite bleeding. The same results for major bleeding incidence were achieved while comparing apixaban to all other anticoagulants (warfarin, rivaroxaban and dabigatran) (6.7 vs. 19.0%, $p = 0.027$). A multivariable Cox regression has shown there is no association with an increased risk of bleeding in NOAC-based treatment when compared to warfarin. What is significant is that a history of bleeding events (HR 3.86, CI 1.41–10.57, $p = 0.009$) and having a deceased kidney donor (HR 2.74, CI 1.16–6.45, $p = 0.021$) were associated with an increased link of bleeding events [47]. Consistently, in a pilot study conducted by Leon, Sabbah et al., NOAC administration in 52 KTRs was investigated and compared to VKA-based treatment in a control group. In terms of bleeding rate, non-vitamin K antagonist oral anticoagulants caused less complications ($p = 0.037$, HR 0.39), with no change in renal function, anemization or graft rejection rates. No thrombotic events have been reported in this study [48]. In a study conducted by Mohammad et al., that included 16 patients divided in two equal groups on NOACs and warfarin, similar observations were made. There were no thromboembolic events, rejection episodes, bleeding or admissions due to NOAC-related adverse events. There were three cases of bleeding in the warfarin group. The most common indication for anticoagulation in both groups was atrial fibrillation (62.5% in group taking NOACs and 50% in group using warfarin, respectively), followed by DVT (37.5% in NOAC group) and valve replacement (25% in warfarin group). In the NOAC group, six patients received rivaroxaban, one patient received dabigatran and one patient received apixaban. Calcineurin inhibitor levels and estimated glomerular filtration rate did not change significantly in the NOAC group ($p = 0.34$ and 0.96, respectively). The study showed that, compared to warfarin, NOACs are well-tolerated and effective for preventing and treating thromboembolic events in KTRs [49]. Those results are however of limited significance due to a small study population. Pfrepper, Herber et al., conducted a study retrospectively and prospectively assessing the safety and efficacy of direct-acting oral anticoagulants (DOAC) in 47 patients who had undergone solid organ transplantation (SOT), 19 of which were KTRs. During the observation period there was no anticoagulation treatment-related death, thromboembolic event or graft rejection reported. A case–control matched assessment showed a significant increase in immunosuppressive agents in the group treated with direct-acting oral anticoagulants ($+3.0 \pm 16.5\%$, $p = 0.004$), but no significant difference in necessity dose adjustments was reported in comparison to patients who did not receive DOACs. Moreover, minor bleeding was reported significantly more often by patients who received rivaroxaban, in comparison to apixaban (70.6 vs. 29.4%, $p = 0.004$), as well as in KTRs if compared to liver transplant recipients, but without statistical significance ($p = 0.063$). In all patients, DOAC concentration remained within the therapeutic range and only a significant increase in apixaban level in patients who took 5 mg twice a day, when compared to the non-transplant controls. [50]. A different comparative approach was applied by Salerno and colleagues in a study assessing the bleeding incidence in SOT recipients who received DOACs but were divided into groups receiving apixaban and dabigatran or rivaroxaban (non-apixaban group), consisting of 70 and 36 patients, respectively. The study showed that apixaban-based anticoagulation

therapy was associated with less bleeding incidents in the non-apixaban group (*p* = 0.034); however, no change in major bleeding events and thromboembolic events was observed between populations (*p* = 0.686 and *p* = 0.515, respectively). There was also no difference among patients who have undergone different SOT [51]. In terms of safety and monitoring of anticoagulation treatment, there are new biomarkers being studied, which may show which patients are at risk of severe complications—most importantly, clinically significant non-major bleeding, major bleeding, thromboembolic events and death. As described by Matusik and colleagues, in patients with NVAF and stage 4 CKD there is a correlation between the concentration of cystatin C, growth differentiation factor-15, high-sensitivity cardiac troponin T and the incidence of clinically significant and major bleeding. Moreover, there is a correlation between decreased plasma fibrin clot permeability and an increased risk of thromboembolic events [52]. However, there are currently no studies carried out on the KTR population.

## 7. Conclusions

Considering the possible adverse effects, anticoagulation treatment in KTRs and patients with CKD is a challenge. As for now, no significant increase in VTE, systemic embolism or bleeding events has been reported in patients who were treated with apixaban, rivaroxaban or dabigatran etexilate. Importantly, current research has provided the best results in patients treated with apixaban in comparison to other DOACs and VKA. Available data prove that NOACs can be used in this group of patients with a level of safety comparable to VKA, but further research is needed to provide clinicians with a clear and complete perspective on the possible lines of treatment, as the recommendations still do not fully support the use of NOACs in KTRs, who receive CNI and mTOR inhibitors. It should be also taken into account that the use of anticoagulation therapy in KTRs should be monitored by experienced teams of clinicians, as some studies on a general population of patients with NVAF show higher levels of safety in patients treated in academic hospitals, where the treatment is introduced in alignment with the latest recommendations [53].

**Author Contributions:** Conceptualization, Z.H.; methodology, M.M. and Z.H.; software, M.M. and Z.H. data curation, A.D.-Ś.; writing—original draft preparation, M.M. and M.S.; writing—review and editing, Z.H. and A.D.-Ś.; visualization, M.M. and M.S.; supervision, A.D.-Ś.; project administration, Z.H.; funding acquisition, A.D.-Ś. All authors have read and agreed to the published version of the manuscript.

**Funding:** This research received no external funding.

**Institutional Review Board Statement:** Not applicable.

**Informed Consent Statement:** Not applicable.

**Conflicts of Interest:** The authors declare no conflict of interest.

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
