# Peer review of "Safety of Non-Vitamin K Antagonist Oral Anticoagulant Treatment in Patients with Chronic Kidney Disease and Kidney Transplant Recipients"

_2673-3943, doi:10.3390/transplantology3030022_

Round 1

Reviewer 1 Report

Thank you for allowing me to review this article. This manuscript by Mlynski and colleagues provides a brief overview of the current literature of the NOAC's in patients with CKD/transplant. The discussion includes current evidence on use in non-CKD population along with import considerations around the CKD population including drug interactions.

It is a well written and nicely organized article. The topics that are chosen for this review are adequate and thoughtful.

Unfortunately, there is a considerable number of articles that are missing in this review. A quick search in Pubmed revealed a number of articles of interest that are not included in this manuscript.

There are many other reviews and meta-analyses that are done in this topic. This manuscript focuses more on a general summary of the topic without any methodological rigor normally used in a review article.

For it to be suitable for publication I would recommend that the authors do a more thorough literature search; include relevant data and having a more methodological rigor normally seen in review articles.

Author Response

The aim of this article is to provide an overview on the topic of safety of oral anticoagulant use in the kidney transplant recipients population.

According to the reviewer’s suggestion, we have included more up-to-date reviews done on this topic:

  1. Leon, J.; Sabbah, L. et al. Efficacy and Safety of Direct Oral Anticoagulants in Kidney Transplantation: A Single-center Pilot Study. 2020, 104(12), 2625-2631.
  2. Mohammad F.Z.; Mahmoud E.S. et al. The Use of Non–Vitamin K Antagonist Oral Anticoagulants in Post-Kidney Transplantation. Single-Center Experience. Transplant Proc. 2021, 52(10), 2918-2922.
  3. Salerno, D.M.; Thornberg, M.E. et al.Less bleeding associated with apixaban versus other direct acting oral anticoagulation in solid organ transplant recipients. Clin Transplant. 2021, 35(12), e14396.

And added a chapter regarding the safety of NOACs regards drug-food interactions of NOACs, according to the suggestion of a different reviewer, based on this review:

  1. Grześk, G.; Rogowicz D. et al. The Clinical Significance of Drug-Food Interactions of Direct Oral Anticoagulants. Int J Mol Sci. 2021, 22(16), 8531.
  2. Matusik, P.T.; Leśniak W.J. et al. Thromboembolism and bleeding in patients with atrial fibrillation and stage 4 chronic kidney disease: impact of biomarkers. Kardiol Pol. 2021, 79(10), 1086-1092.

Thank you for allowing me to review this article. This manuscript by Mlynski and colleagues provides a brief overview of the current literature of the NOAC's in patients with CKD/transplant. The discussion includes current evidence on use in non-CKD population along with import considerations around the CKD population including drug interactions.

It is a well written and nicely organized article. The topics that are chosen for this review are adequate and thoughtful.

Unfortunately, there is a considerable number of articles that are missing in this review. A quick search in Pubmed revealed a number of articles of interest that are not included in this manuscript.

There are many other reviews and meta-analyses that are done in this topic. This manuscript focuses more on a general summary of the topic without any methodological rigor normally used in a review article.

For it to be suitable for publication I would recommend that the authors do a more thorough literature search; include relevant data and having a more methodological rigor normally seen in review articles.

Reviewer 2 Report

Very interesting review.
We need more studies that give us knowlegde about this topic.
The dosage ajustment is very dificult in kidney patients, not only for the renal clerance otherwise of the interactions with other drugs. 
I think the table 1 when you say "Dosage in CKD" you have to aclare the eGFR, becuse is not the same in all the stages.

Author Response

According to the reviewer’s suggestion we have included more information in table 1 regarding the specific dosage of novel oral anticoagulants at different eGFR levels in patients.

Very interesting review.
We need more studies that give us knowledge about this topic.
The dosage adjustment is very difficult in kidney patients, not only for the renal clearance otherwise of the interactions with other drugs. 
I think the table 1 when you say "Dosage in CKD" you have to clarify the eGFR, because is not the same in all the stages.

Reviewer 3 Report

The presented article is a review of a clinically important topic, that is safe application of non-vitamin K antagonist oral anticoagulant in treatment of patients with chronic kidney disease and of kidney transplant recipients. The paper provides a thorough insight into different drugs, their interactions and safety in this specific population. Moreover, the paper is very well written and includes all the recent information regarding the described drugs. This is thanks to good selection of up-to-date source materials. I believe that the paper is worth publication, mainly because it provides comprehensive knowledge in a concise form. It may be very helpful for practitioners.

Author Response

We are grateful for this positive review of our article. According to the suggestion of other reviewers, we have included more studies on the topic of novel oral anticoagulants usage in kidney transplant recipient population:

  1. Leon, J.; Sabbah, L. et al. Efficacy and Safety of Direct Oral Anticoagulants in Kidney Transplantation: A Single-center Pilot Study. 2020, 104(12), 2625-2631.
  2. Mohammad F.Z.; Mahmoud E.S. et al. The Use of Non–Vitamin K Antagonist Oral Anticoagulants in Post-Kidney Transplantation. Single-Center Experience. Transplant Proc. 2021, 52(10), 2918-2922.
  3. Salerno, D.M.; Thornberg, M.E. et al.Less bleeding associated with apixaban versus other direct acting oral anticoagulation in solid organ transplant recipients. Clin Transplant. 2021, 35(12), e14396.
  4. Matusik, P.T.; Leśniak W.J. et al. Thromboembolism and bleeding in patients with atrial fibrillation and stage 4 chronic kidney disease: impact of biomarkers. Kardiol Pol. 2021, 79(10), 1086-1092.

And added a chapter regarding the safety of NOACs regards drug-food interactions of NOACs, according to the suggestion of a different reviewer, based on this review:

  1. Grześk, G.; Rogowicz D. et al. The Clinical Significance of Drug-Food Interactions of Direct Oral Anticoagulants. Int J Mol Sci. 2021, 22(16), 8531.

The presented article is a review of a clinically important topic, that is safe application of non-vitamin K antagonist oral anticoagulant in treatment of patients with chronic kidney disease and of kidney transplant recipients. The paper provides a thorough insight into different drugs, their interactions and safety in this specific population. Moreover, the paper is very well written and includes all the recent information regarding the described drugs. This is thanks to good selection of up-to-date source materials. I believe that the paper is worth publication, mainly because it provides comprehensive knowledge in a concise form. It may be very helpful for practitioners.

Reviewer 4 Report

The authors undertook the difficult task of reviewing the clinical safety assessment of patients with chronic kidney disease. The work contains current references to current works, but I have the impression that the authors focused mainly on the results of large pre-registration studies. The problem of safety in patients with impaired renal function is particularly important, because on the one hand we strive to implement the best treatment ensuring the highest possible effectiveness, and on the other hand such therapy must be simply safe.
One of the most important methods of managing the risk of haemorrhagic complications in these patients is the implementation of monitored therapy carried out under the supervision of experienced centers providing therapy with drug monitoring. The requirement of extensive experience results from the lack of concentration standards, therefore, based on the desired values ​​and experience, it is possible to safely conduct therapy in clinically difficult situations.
I believe that the work is interesting, but in order for it to be of greater value and to be published, it should be extended with a few elements:
1. there are different ways of assessing, but the fact of the progression of chronic kidney disease has been greatly neglected, please emphasize this aspect in particular and compare the progression to end-stage renal disease.
2.DOAC monitored therapy in some centers started already in the fall of 2012, hence there are papers describing 8 years of experiments, as well as better defining the real concentration ranges in the treated patients. I believe that it is necessary to express an opinion and take these data into account!
3. interactions between immunosuppressants and many others are quite well described in the EHRA guidelines, there is another large group of drugs whose use is associated with the presence of pharmacokinetic and pharmacodynamic interactions that may significantly affect the safety of pharmacotherapy. An example is the use of nintedanib and DOAC and the management of the risk of bleeding complications with the consideration of TDM.
4. Following the introduction of the DOAC, it was possible to find information that there was no significant interaction with the diet. Several examples of interactions have been published in the EHRA 2021 Handbook, but papers describing several dozen DOAC interactions with diet have already been published. This is an extremely important aspect of herd therapy and must be taken into account at work.

I believe that the paper assesses an extremely important aspect of DOAC therapy, therefore it is worth publishing it, but only after taking into account the mentioned aspects.

Author Response

In accordance with reviewer’s opinion, we have addressed the suggestion listed below:

  1. The main aim of this article is to describe the up-to-date knowledge on direct oral anticoagulant usage in kidney transplant recipients receiving immunosuppression therapy. The aspect of novel oral anticoagulant-based treatment in patients with chronic kidney disease is mentioned as a complimentary note in order to provide a more broad overview on the clinical management of nephrological patients requiring anticoagulant therapy.
  2. The studies referred to by the reviewer included research focusing on determining the plasma therapeutic drug concentrations. We believe that, due to a stable pharmacokinetic and pharmacodynamic profile of this group of drugs, it is most important to monitor the physiological effects of DOAC-based therapy. We mentioned information on treatment monitoring with anti-Xa activity or dTT for apixaban and rivaroxaban or dabigatran respectively. We have included more transparent information on that topic as well as a paragraph regarding the correlation between plasma drug concentration and therapeutic effects.
  3. The article covers mainly the intricacies of DOAC-based treatment in kidney transplant recipients, who receive immunosuppressive agents, most common of which have been listed in the article – importantly, calcineurin inhibitors and their interactions, specifically with dabigatran. The description of other drug-to-drug interactions, not directly regarding the KTRs population, may appear confusing, given the primary aim of this article. However, we agree it is crucial for clinicians to take other interactions into consideration, especially in such a specific population.
  4. We agree that drug-food and drug-herb interactions may pose a significant hazard during DOAC-based therapy, especially in patients already at risk of developing drug-to-drug interactions. Thus, we have included data published on the topic of DOAC interactions with food and dietary supplements in the latest studies.

The authors undertook the difficult task of reviewing the clinical safety assessment of patients with chronic kidney disease. The work contains current references to current works, but I have the impression that the authors focused mainly on the results of large pre-registration studies. The problem of safety in patients with impaired renal function is particularly important, because on the one hand we strive to implement the best treatment ensuring the highest possible effectiveness, and on the other hand such therapy must be simply safe.
One of the most important methods of managing the risk of haemorrhagic complications in these patients is the implementation of monitored therapy carried out under the supervision of experienced centers providing therapy with drug monitoring. The requirement of extensive experience results from the lack of concentration standards, therefore, based on the desired values ​​and experience, it is possible to safely conduct therapy in clinically difficult situations.
I believe that the work is interesting, but in order for it to be of greater value and to be published, it should be extended with a few elements:
1. there are different ways of assessing, but the fact of the progression of chronic kidney disease has been greatly neglected, please emphasize this aspect in particular and compare the progression to end-stage renal disease.
2.DOAC monitored therapy in some centers started already in the fall of 2012, hence there are papers describing 8 years of experiments, as well as better defining the real concentration ranges in the treated patients. I believe that it is necessary to express an opinion and take these data into account!
3. interactions between immunosuppressants and many others are quite well described in the EHRA guidelines, there is another large group of drugs whose use is associated with the presence of pharmacokinetic and pharmacodynamic interactions that may significantly affect the safety of pharmacotherapy. An example is the use of nintedanib and DOAC and the management of the risk of bleeding complications with the consideration of TDM.
4. Following the introduction of the DOAC, it was possible to find information that there was no significant interaction with the diet. Several examples of interactions have been published in the EHRA 2021 Handbook, but papers describing several dozen DOAC interactions with diet have already been published. This is an extremely important aspect of herd therapy and must be taken into account at work.

I believe that the paper assesses an extremely important aspect of DOAC therapy, therefore it is worth publishing it, but only after taking into account the mentioned aspects.

Round 2

Reviewer 4 Report

I suggest to accept without additional amendments